# Association between dietary contribution of ultra-processed foods and urinary concentrations of phthalates and bisphenol in a nationally representative sample of the US population aged 6 years and older

Eurídice Martínez Steele[1,2]*, Neha Khandpur[1,2,3], Maria Laura da Costa Louzada[1,2], Carlos Augusto Monteiro[1,2]

1 Department of Nutrition, School of Public Health, University of São Paulo, São Paulo, Brazil, 2 Center for Epidemiological Studies in Health and Nutrition, University of São Paulo, São Paulo, Brazil, 3 Department of Nutrition, Harvard T. H. Chan School of Public Health, Boston, Massachusetts, United States of America

* emar_steele@hotmail.com

**Data Availability Statement:** All analyses used publicly available datasets downloadable from

## Abstract

Ultra-processed food consumption has been associated with several health outcomes such as obesity, hypertension, cardiovascular disease and cancer. The deleterious nutrient profile of these products, and the presence of food additives, neoformed contaminants and contact materials such as phthalates and bisphenol may be some of the potential pathways through which ultra-processed food influences disease outcomes. The aim of this study was to examine the association between dietary contribution of ultra-processed foods and urinary biomarker concentrations of parent compounds or their metabolites including Di(2-ethylhexyl) phthalate (ΣDEHP), Di-isononyl phthalate (ΣDiNP), Monocarboxynonyl phthalate (mCNP), Mono (3-carboxypropyl) phthalate (mCPP), Monobenzyl phthalate (mBzP), Bisphenol A (BPA), Bisphenol F (BPF) and Bisphenol S (BPS), in the US. Participants from the cross-sectional 2009–2016 National Health and Nutrition Examination Survey, aged 6+ years, with urinary measures and with one 24-hour dietary recall were included in the study. Ultra-processed foods were identified based on the NOVA classification system, a four-group food classification based on the extent and purpose of industrial food processing. Linear regression was used to compare average urinary creatinine-standardized concentrations across quintiles of energy contribution of ultra-processed foods. Models incorporated survey sample weights and were adjusted for different sociodemographic and life-style variables. Adjusted geometric means of ΣDiNP, mCNP, mCPP, mBzP and BPF increased monotonically from the lowest to the highest quintile of ultra-processed food consumption. As both phthalates/bisphenol and ultra-processed foods have been previously associated with insulin resistance, diabetes, general/abdominal obesity and hypertension, our results suggest the possibility of contact materials in ultra-processed foods as one link between ultra-processed food and these health outcomes. Future studies could confirm findings and further explore these mechanisms of action.

NHANES website (<https://wwwn.cdc.gov/nchs/nhanes/Default.aspx>).

**Funding:** This research received funding from Fundação de Amparo à Pesquisa do Estado de São Paulo (Processo n° 2015/14900-9) to CAM and from Fundação de Amparo à Pesquisa do Estado de São Paulo (Processo FAPESP n° 2018/17972-9) to EMS.

**Competing interests:** The authors have declared that no competing interests exist.

**Abbreviations:** BBzP, Benzylbutyl phthalate; BPA, Bisphenol A; BPF, Bisphenol F; BPS, Bisphenol S; DEHP, Di(2-ethylhexyl) phthalate; DiDP, Di-isodecyl phthalate; DiNP, Di-isononyl phthalate; DOP/DnOP, Di-n-octyl phthalate.

## Introduction

Ultra-processed foods are defined by NOVA (not an acronym) classification, as industrial formulations of food-derived substances (such as oils, fats, sugars, starch, protein isolates) that contain little or no whole food and often include flavorings, colorings, emulsifiers and other cosmetic additives [1]. Over the past decades, the consumption of ultra-processed foods has increased worldwide [2–8]. Prospective studies have linked ultra-processed food intake with a higher risk of overweight, obesity [9, 10], hypertension [11], dyslipidaemia [12], overall and breast cancer [13], cardiovascular diseases [14], diabetes [15] and all- cause mortality [16–18].

Several mechanisms may potentially explain these associations. Ultra-processed foods have a higher content in total fat, saturated fat, added sugar, energy density, and salt, together with a lower fibre, vitamin and mineral density, as compared to non-ultra-processed foods. Their consumption results in an overall deterioration of the nutritional quality of the diet [1, 19]. The convenience and hyperpalatability of ultra-processed foods, simultaneously lowers consumption of healthy non-ultra-processed foods such as fruit and vegetables [19]. Ultra-processed foods may also affect glycaemic responses and satiety [20] and create a gut microbial environment that promotes inflammatory disease [21]. Cosmetic additives frequently added to ultra-processed foods (such as glutamates, emulsifiers, sulfites and carrageenan) or several compounds that are neoformed during their processing (such as acrylamide or acrolein) could also promote disease [14]. A recent inpatient ad libitum cross-over randomized controlled trial conducted by the US National Institute of Health demonstrated that individuals consumed 508 more kcal/day and gained an average of 0.8 kg of weight during the 2-week ultra-processed diet ($>$ 80% energy from ultra-processed foods) and lost 0.9 kg during the 2-week non-ultra-processed diet. The fact that diets were matched for total calories, macronutrients and fiber, suggests that mechanisms other than the dietary nutrient profile, like quicker eating time or reduced signs of satiety, might explain these results [22].

While not directly related with the food per se, the packaging of ultra-processed foods might also help explain the health effects of these products [14]. Ultra-processed foods are frequently packaged in materials that are a source of endocrine disrupting chemicals such as phthalates and bisphenol, associated with adverse health outcomes especially in pregnancy [23, 24]. A large body of cross-sectional studies have specifically linked Bisphenol A (BPA) exposure with higher risk of diabetes, general/abdominal obesity, and hypertension [25] and phthalates with diabetes [26] and insulin resistance [27].

Though contained in many consumer products, food is a ubiquitous source of phthalates and bisphenols attributed mainly to food production, processing, and packaging practices; food storage conditions and, also animal feeding practices [24]. These chemicals are not bound to the polymer matrix chemically and are known to migrate from food contact materials (plastics, paper, metal, glass, and printing inks) that protect food from physical damage and microbial spoilage [24]. The long-shelf life and ready-to-eat characteristics of ultra-processed foods entails that these substances are likely to leach into the food product, making ultra-processed foods a potential delivery vehicle for phthalates and bisphenols in humans. This leakage could be more severe in ready-to-eat or take-away food often heated or served warm in paper, cardboard or plastic containers [24, 28, 29].

The objective of our study was to examine the association between dietary contribution of ultra-processed food and exposure to Di(2-ethylhexyl) ($\Sigma$DEHP), Di-isononyl ($\Sigma$DiNP), Monocarboxynonyl (mCNP), Mono(3-carboxypropyl) (mCPP) and Monobenzyl (mBzP) phthalates, and Bisphenol A, F and S (BPA, BPF and BPS, respectively), in a US population aged 6 years and older.

Very few studies have explored this topic. To our knowledge, only one other study has assessed the link between ultra-processed food consumption and phthalates/bisphenol [30]. The authors of this study found a positive association between ultra-processed foods and urinary concentrations of Monocarboxynonyl (mCNP), Mono(3-carboxypropyl) (mCPP), and mono-(carboxyisoctyl) (MCOP) but not mono-benzyl (MBzP), Di(2-ethylhexyl) (ΣDEHP), or bisphenols. This study included data from NHANES cycle 2013–14 and was most likely underpowered to detect associations. The current study addresses this gap by including data from 2009 to 2016. We also examined the departure from linear relationship between percent of calories from ultra-processed foods and urinary concentrations of phthalate or bisphenol biomarkers. In addition, several sensitivity analyses were carried out to test the robustness of associations.

## Material and methods

### Data source, population and sampling

We used nationally representative data from National Health and Nutrition Examination Survey (NHANES) 2009–2016 (four 2-year cycles). NHANES is a continuous, nationally representative, cross-sectional survey of non-institutionalized, civilian US residents conducted by The Centers for Disease Control and Prevention [31]. Participants were recruited using a four-stage sample design based on the selection of counties, blocks, households, and the number of people within households.

The survey included an interview conducted in the home and a subsequent health examination performed at a mobile examination center (MEC) that included blood and urine collection. All NHANES participants who were examined at MECs were eligible for two 24-hour dietary recall interviews: the first one collected in-person in the MEC [32] and the second by telephone, 3 to 10 days later [33]. Dietary interviews were conducted by trained interviewers using the validated [34–36] US Department of Agriculture Automated Multiple-Pass Method [37]. Proxy-assisted interviews were conducted with children 6–11 years old; participants $\geq$ 12 years old completed the dietary interview for themselves.

Our analytical sample comprised individuals aged 6 years or older (urinary concentrations of phthalate or bisphenol biomarkers were not measured in < 6 year old children), who provided a urine sample for phthalate or bisphenol analysis, completed a 24-hr dietary recall survey and had complete information on all variables of interest. This, resulted in a final sample size of 9,416 participants for phthalate analysis and 9,420 for Bisphenol A. The final sample size for Bisphenol F and S analyses was 4,655, as these 2 urinary concentrations were only measured in cycles 2013–2016 [38] (Fig 1).

The National Center for Health Statistics Research Ethics Review Board approved the study protocol. All participants provided written informed consent; parents or guardians provided consent for participants < 18 years of age.

### Urinary chemical measurement

Due to their quick metabolism and consequent short half-lives (<24 h), exposures to phthalates and bisphenols are best characterized in urine (compared with blood) [39, 40]. Our study focused on the phthalate and bisphenol biomarkers (expressed in ng/mL) measured in all 4 studied cycles including Mono(2-ethylhexyl) (mEHP), Mono(2-ethyl-5-hydroxyhexyl) (mEHHP), Mono(2-ethyl-5-oxohexyl) (mEOHP), Mono(2-ethyl-5-carboxypentyl) (mECPP), Mono-isononyl (mNP/miNP), Monocarboxyoctyl (mCOP), Monocarboxynonyl (mCNP), Mono(3-carboxypropyl) (mCPP), Monobenzyl (mBzP), Monoethyl (mEP), Mono-n-butyl

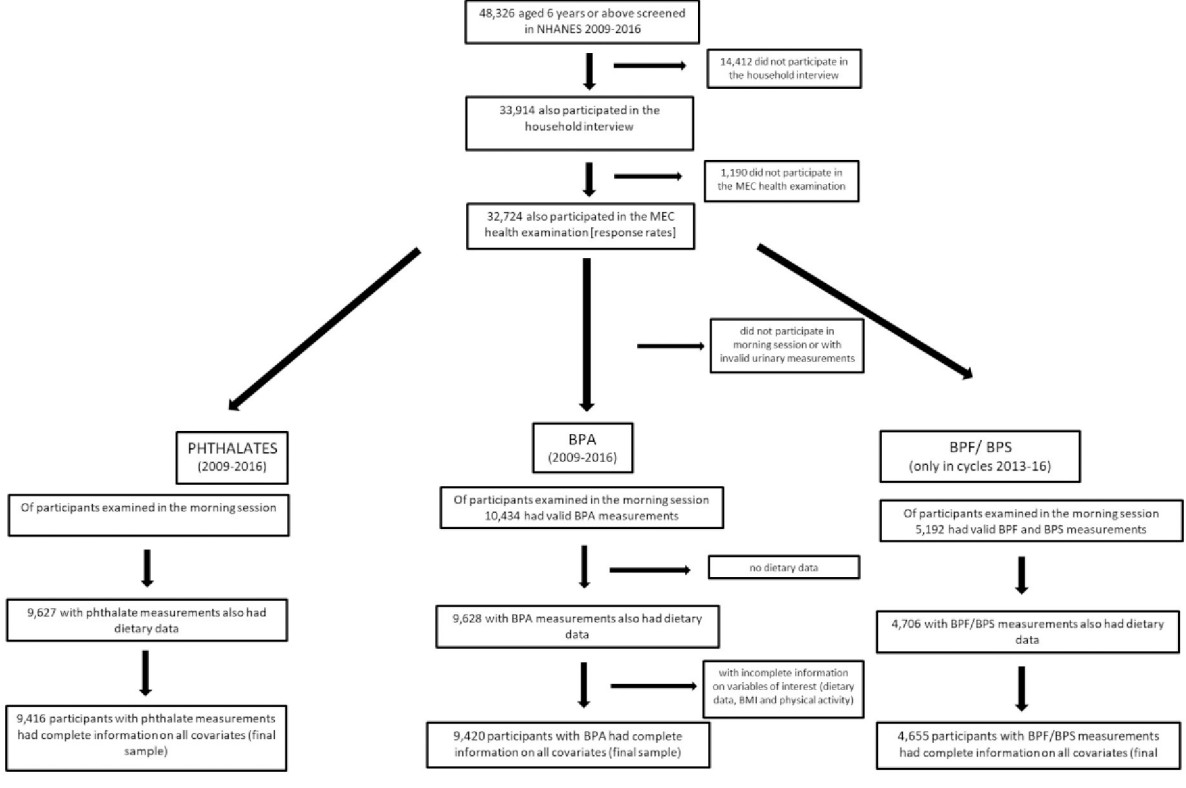

**Fig 1. Study flowchart.**

(mnBP), Mono-isobutyl (miBP), Bisphenol A (BPA) and its replacements Bisphenol S (BPS) and Bisphenol F (BPF) (S1 Table).

Urine specimens were collected in spot urine samples at the MEC and processed, stored under appropriate frozen (–20˚C) conditions, and shipped to the Division of Laboratory Sciences, National Center for Environmental Health, Centers for Disease Control and Prevention for analysis. Chemical analytes were quantified in urine using solid-phase extraction coupled online with high-performance liquid chromatography and tandem mass spectrometry and expressed as wet weights (ng/mL) [41, 42]. The limits of detection (LOD) ranged from 0.2 to 1.2 ng/mL for the phthalates [41] and from 0.1 to 0.4 ng/mL for bisphenols [42]. Where LOD varied across study cycles, we assumed the maximal LOD for each phthalate and bisphenol in our analysis to facilitate aggregation of data across study cycles [43].

For the sample of 9,416, 2858 individuals were below the lower detection limit (LLOD) for mEHP (0.8 ng/mL), 34 individuals for mEHHP (0.4 ng/mL), 40 individuals for mEOHP (0.2 ng/mL), 16 individuals for mECPP (0.4 ng/mL), 5533 for mNP/miNP (0.9 ng/mL), 20 for mCOP (0.3 ng/mL), 121 for mCNP (0.2 ng/mL), 808 for mCPP (0.4 ng/mL), 158 for mBzP (0.3 ng/mL), 18 for mEP (1.2 ng/mL), 200 for mBP/mnBP (0.4 ng/mL) and 130 for miBP (0.8 ng/mL). For the sample of 9,420, 637 individuals were below the LLOD for BPA (0.4 ng/mL). For the sample of 4655, 466 individuals were below the LLOD for BPS (0.1 ng/mL) and 2,093 for BPF (0.2 ng/mL). In NHANES, urinary phthalate and bisphenol measurements below the limits of detection of the used method were replaced with $1/\sqrt{2}$ fraction of the detection limit. Individual concentrations (expressed in ng/mL) were rescaled in ηmol/mL by dividing each one by its molar mass. We calculated molar sums, representing classes of chemicals or parent

compounds, by summing individual metabolite concentrations [44]: ΣDEHP (sum of di (2-ethylhexyl) phthalate metabolites: MEHP, MEHHP, MEOHP, and MECPP), and ΣDiNP (sum of Di-isononyl phthalate metabolites: mNP/miNP, and mCOP). In order to correct for urine dilution, urinary concentrations were normalized by urinary creatinine (and expressed in ηmol/g creatinine) [45]. This was done by dividing each individual concentration value (expressed in nmol/mL) by the corresponding urinary creatinine value (expressed in g/mL). Creatinine was measured using Beckman Synchron CX3 Clinical Analyser at the University of Minnesota [46].

## Food classification according to level of processing

During the dietary interview, participants were prompted to list all foods and beverages consumed the day prior to the interview (in a 24-hr period, from midnight to midnight). All recorded food items (Food Codes) were classified according to NOVA, a food classification based on the extent and purpose of industrial food processing. NOVA includes 4 groups: "unprocessed or minimally processed foods" (such as fresh, dry or frozen fruits or vegetables; packaged grains and pulses; grits, flakes or flours made from corn, wheat or cassava; pasta, fresh or dry, made from flours and water; eggs; fresh or frozen meat and fish and fresh or pasteurized milk); "processed culinary ingredients" (including sugar, oils, fats, salt, and other substances extracted from foods and used in kitchens to season and cook unprocessed or minimally processed foods and to make culinary preparations), "processed foods" (including canned foods, sugar-coated dry fruits, salted meat products, cheeses and freshly made unpackaged breads, and other ready-to-consume products manufactured with the addition of salt or sugar or other substances of culinary use to unprocessed or minimally processed foods), and "ultra-processed foods".

The NOVA group of ultra-processed foods of particular interest in this study, includes soft drinks, sweet or savory packaged snacks, confectionery and industrialized desserts, mass-produced packaged breads and buns, poultry and fish nuggets and other reconstituted meat products, instant noodles and soups, and many other ready-to-consume formulations of several ingredients. Besides salt, sugar, oils, and fats, these ingredients include food substances not commonly used in culinary preparations, such as modified starches, hydrogenated oils, protein isolates and classes of additives whose purpose is to imitate sensorial qualities of unprocessed or minimally processed foods and their culinary preparations, or to disguise undesirable qualities of the final product. These additives include colorants, flavorings, non-sugar sweeteners, emulsifiers, humectants, sequestrants, and firming, bulking, de-foaming, anti-caking and glazing agents. Unprocessed or minimally processed foods represent a small proportion of or are even absent from the list of ingredients of ultra-processed foods. A detailed definition of each NOVA food group and examples of food items classified in each group are shown elsewhere [47].

For all food items (Food Codes) judged to be a handmade recipe, the classification was applied to the underlying ingredients (Standard Reference Codes -SR Codes-) obtained from the USDA Food and Nutrient Database for Dietary Studies (FNDDS) [FNDDS] as further explained in previously published papers [48, 49]. Food items were sorted into mutually exclusive food subgroups within unprocessed or minimally processed foods (n = 11), processed culinary ingredients (n = 4), processed foods (n = 4) and ultra-processed foods (n = 18) [48].

To account for the propensity of phthalates and bisphenols to be concentrated in ready-to-eat or take-away food often heated or served warm in paper, cardboard or plastic containers [24, 28, 29], ultra-processed food subgroups were sorted into two groups: (1) Ready-to-heat/ Frozen meals (Frozen and shelf-stable plate meals; Pizza; French fries and other potato

products; Sandwiches and hamburgers on bun); (2) Other ultra-processed foods (Breads; Cakes, cookies and pies; Salty snacks; Soft drinks, carbonated; Fruit drinks; Breakfast cereals; Sauces, dressings and gravies; Reconstituted meat or fish products; Sweet snacks; Ice cream and ice pops; Milk-based drinks; Desserts; Instant and canned soups; Other ultra-processed food).

## Dietary assessment

USDA's Food and Nutrient Database for Dietary Studies 5.0, 2011–2012, 2013–2014 and 2015–2016 [50] were used to code dietary intake data and to calculate Food Code energy and total fat energy intakes (kcal). For handmade recipes, we calculated the underlying ingredient (SR Code) energy and total fat values using variables from both FNDDS databases [50] and USDA National Nutrient Database for Standard Reference, Release 24, 26 and 28 [51]. The dietary recall interview also asked about the source from which each food was obtained. Fast food was defined as food obtained from restaurants without waiter/waitress service, or from pizza restaurants regardless of waiter/ waitress service.

For these analyses, we estimated 24-hr total energy intake, and energy intakes derived from ultra-processed foods, from total fat, from fat in ultra-processed foods, and from fast food, for every participant. We additionally estimated 24-hr total energy intake derived from both Ready-to-heat and Other ultra-processed food NOVA subgroups. The gram intake derived from ultra-processed foods was also estimated for sensitivity analysis.

## Covariates

Potential confounders were identified from the literature [24, 30]. Socio-demographic covariates included sex, age, race/ethnicity, family income and cycle. Age was grouped into three categories (6–11 years, 12–19 years, 20 years of age and over) [31]. Race/ethnicity was categorized as Mexican- American, Other Hispanic, Non-Hispanic White, Non-Hispanic Black and Other Races including Multi-Racial. With respect to family income, ratio of family income to poverty was established and categorized based on Supplemental Nutrition Assistance Program (SNAP) eligibility as 0.00–1.30, >1.30–3.50, and 3.50 and above [31]. Cycle included the following categories: 2009–10, 2011–12, 2013–14 and 2015–16.

BMI was calculated by dividing measured weight by height squared (kg/m2) [52] and used as a categorical covariate (underweight, normo-weight, overweight and obesity). Among adults, BMI values of <20 were classified as underweight, ≥25 as overweight and ≥30 kg/m2 as obesity according to World Health Organization criteria [53]. Among children, cutoff criteria were based on the Centers for Disease Control and Prevention's sex-specific 2000 BMI-for-age-sex growth charts for the United States: Underweight (BMI < 5th percentile), Normal weight (BMI 5th to < 85th percentiles), Overweight (BMI 85th to < 95th percentiles) and Obese (BMI ≥ 95th percentile) [54].

Physical activity was categorized into three intensity levels–light (<150 minutes per week of moderate intensity equivalent activity), moderate (150 to 300 minutes per week of moderate-intensity equivalent activity) and vigorous (>300 minutes per week of moderate- intensity equivalent activity) [55]. In individuals 12 + years of age, MET (metabolic equivalent of task)-minutes per week were calculated based on reported frequency and duration of physical activity in a typical week. In children<12 years of age, minutes per week were calculated based on response to question "Days physically active during past week (at least 60 minutes)".

Smoking status was categorized as current smoker and non-smoker. Energy intake above recommended levels was coded as yes/ no according to sex–age–physical activity levels [56]. Total fat intake (% of total energy intake) was used as continuous or categorized as >30% (yes/

no) [57]. Fat in ultra-processed food derived total energy intake (% of total energy intake) was categorized as > median value (18.7%) (yes/no) and into quintiles. Fast food intake (% of total energy intake) was used as continuous.

## Data analysis

All available day 1 dietary intake data for each participant were utilized. First, we evaluated the mean dietary contribution of ultra-processed food (% of total energy) and urinary phthalate and bisphenol concentrations, overall and across socio-demographic, life- style and dietary characteristics of respondents using linear regression. Test of linear trend was performed for ordinal variables and Wald test with Bonferroni inequality adjustment for multiple comparisons was used for non-ordinal categorical variables or in the absence of a statistically significant linear trend. As urinary biomarker concentrations (both in ηmol/ ml and standardized in ηmol/g creatinine) had skewed distributions, these variables were log transformed (using natural logarithms) and geometric means were presented.

The average biomarker urinary concentrations were compared across quintiles of the dietary contribution of ultra-processed foods (% of total energy intake) using linear regression models. For each phthalate and bisphenol biomarker, four models were explored: 1) crude (in ηmol/ml); 2) standardized (in ηmol/g creatinine); 3) standardized (in ηmol/g creatinine) and adjusted for socio-demographic variables (sex, age group, race/ethnicity, ratio of family income to poverty, cycle); and 4) standardized (in ηmol/g creatinine) and adjusted for socio-demographic variables (sex, age group, race/ethnicity, ratio of family income to poverty, cycle), energy intake above recommended (yes, no), BMI (categorical), physical activity (categorical) and current smoking (yes, no). From these regression models, we estimated: a) geometric means of urinary chemical concentrations across quintiles of the energy contribution of ultra-processed foods as $e^{(constant + \beta)}$, where β are each of the estimated regression coefficients. On the basis of the multivariable regression models, margins were estimated at the means of all covariates; b) percent difference in urinary chemical concentrations comparing first and fifth quintile of the energy contribution of ultra-processed foods as $(e^{(\beta 4)} - 1) \times 100\%$ with 95% CIs estimated as $(e^{(\beta 4 \pm critical\ value \times SE)} - 1) \times 100\%$, where β4 and SE are the estimated regression coefficient and standard error for the fifth quintile, respectively. Tests of linear trend were also performed to evaluate the effect of quintiles as a single continuous variable.

Thereafter, we used the restricted cubic spline in the multivariable linear regression models with five knots (5th, 27.5th, 50th, 72.5th, and 95th) following Harrell´s recommendations [58], to examine the shape of the dose-response relationship curve between percent of calorie from ultra-processed foods and urinary chemical concentrations [59].

To test the robustness of the associations, the following sensitivity tests were performed:

1. Using % of total gram intake of ultra-processed foods instead of % of total energy intake.

2. Previous studies have suggested that foods high in fat may be more contaminated by phthalates and BPA that are more lipophilic [60, 61]. For this reason and to test the robustness of the associations, we conducted sensitivity analyses (using the multivariable socio-demographic and life-style adjusted model) also adjusting for total fat intake (% of total energy intake) (continuous). We also carried out fully adjusted secondary analysis using as exposure variable the quintiles of fat in ultra-processed food derived total energy intake (% of total energy intake).

3. As studies have suggested that fast food may be a unique dietary source of ΣDiNP and mBzP [62, 63], we carried out sensitivity analysis also adjusting for fast food intake (% of total energy intake).

4. Because of lack of consensus on the most appropriate method to adjust for urinary dilution, the use of different methods for urinary dilution adjustment has been recommended [40]. For this reason, sensitivity analyses were also carried out adjusting for creatinine concentration (milligrams per deciliter) while using crude concentration measures (ηmol/mL) as suggested by Barr et al. [64].

Effect modification by sex, age group and data collection cycle were tested by including a one-by-one multiplicative interaction term (tested both as continuous and as dummy variable) in the multivariable socio-demographic and life-style adjusted model. Analyses were stratified according to statistically significant interaction variables and cycle.

We also examined the association between dietary contribution of Ready-to-heat and Other ultra-processed foods (each categorized into tertiles) and phthalate/bisphenol levels using adjusted models. Finally, we performed secondary analyses to test the association between quintiles of dietary contribution of each of the remaining three NOVA groups (minimally processed foods, processed culinary ingredients and processed foods) and urinary concentrations.

NHANES survey sample weights were used in all analyses to account for differential probabilities of selection for the individual domains, nonresponse to survey instruments, and differences between the final sample and the total US population. The Taylor series linearization variance approximation procedure was used for variance estimation in all analysis to account for the complex sample design and the sample weights [31]. Because we combined four survey cycles, new sample weights were calculated for each participant according to the analytical guidelines [31]. Statistical hypotheses were tested using a two-tailed p<0.05 level of significance. Data were analyzed using Stata statistical software package version 14.

## Results

Overall dietary contribution of ultra-processed foods was 58% and decreased with age and cycle. Intakes were higher among non-Hispanic white and black (and lower among other race) and lower among those in the highest income level. Ultra-processed food consumption varied according to BMI status and was higher among smokers, and among individuals with energy and total fat intake above recommended levels (Table 1).

As can be seen in Table 1, phthalate/bisphenol concentrations were higher among women (except for BPS), decreased with age (except BPF which did not change and BPS which increased with age) and cycle (except for BPS) and varied across race/ethnicities (except for BPA) and income levels (except for BPF). Concentrations of some biomarkers also varied according to BMI status (decreased with BMI for ΣDEHP, mCPP and mBzP), physical activity (higher among middle physical activity level for ΣDEHP, ΣDiNP, mCNP and mCPP) and smoking status (higher among smokers for BPA, BPF, BPS and mBzP, and higher among non-smokers in remaining chemicals). For some biomarkers, levels varied according to excess total energy intake (positive association for ΣDEHP, mCNP and mCPP) and excess total fat intakes (positive association for ΣDiNP and mCNP, and inverse for BPS), and fat in ultra-processed foods derived total energy intake (positive association for all biomarkers, except for BPS which was inverse and ΣDEHP with no association).

Fully adjusted models showed a positive association between ultra-processed food quintiles and ΣDiNP, mCNP, mCPP, mBzP, and BPF concentration levels. Conversely, a lack of association was observed for ΣDEHP and BPA, and an inverse association for BPS (Table 2). Compared to the lowest ultra-processed food consumers (first quintile), the highest quintile had 23.4% (95% CI: 7.9% to 41.2%) higher levels of ΣDiNP, 14.6% (95% CI: 4.4% to 25.8%) higher levels of mCNP, 11.5% (95% CI: 0.2% to 24.1%) higher levels of mCPP, 10.7% (95% CI: -0.6%

**Table 1. Dietary contribution of ultra-processed foods and phthalate/bisphenol standardized levels (ηmol/g creatinine) according to characteristics of respondents.** US population aged 6 and above (NHANES 2009–2016).

| | | Dietary contribution of ultra-processed foods (% total energy intake) (n = 9,416) | PHTHALATES (ηmol/g creatinine) | | | | | BISPHENOL (ηmol/g creatinine)[c] | | |
|---|---|---|---|---|---|---|---|---|---|---|
| | | | ΣDEHP (n = 9,416) | ΣDiNP (n = 9,416) | mCNP (n = 9,416) | mCPP (n = 9,416) | mBzP (n = 9,416) | BPA (n = 9,420) | BPF (n = 4,655) | BPS (n = 4,655) |
| | | mean (SE) | GM[a] (GSE) | GM (GSE) | GM (GSE) | GM (GSE) | GM (GSE) | GM (GSE) | GM (GSE) | GM (GSE) |
| **Gender** | Men | 58.4 (0.4) | 87.8 (1.0) | 49.7 (1.0) | 7.1 (1.0) | 8.5 (1.0) | 17.5 (1.0) | 5.9 (1.0) | 2.1 (1.1) | 1.6 (1.1) |
| | Women | 58.2 (0.5) | 103.6 (1.0)[£] | 56.5 (1.0)[£] | 8.0 (1.0)[£] | 9.3 (1.0)[£] | 22.2 (1.0)[£] | 7.0 (1.0)[£] | 2.4 (1.0)[£] | 1.9 (1.0) |
| **Age groups (years)** | 6 to 11 | 68.2 (0.5) | 176.5 (1.0) | 78.9 (1.1) | 11.7 (1.0) | 16.6 (1.0) | 50.8 (1.0) | 8.8 (1.0) | 2.3 (1.1)[A] | 1.9 (1.0) |
| | 12 to 19 | 66.9 (0.7) | 90.5 (1.0) | 52.1 (1.1) | 6.8 (1.0) | 8.4 (1.1) | 23.8 (1.0) | 5.6 (1.0) | 2.0 (1.1)[A] | 1.2 (1.0) |
| | 20 or above | 55.9 (0.4)* | 90.3 (1.0)* | 51.0 (1.0)* | 7.3 (1.0)* | 8.4 (1.0)* | 17.5 (1.0)* | 6.4 (1.0)* | 2.3 (1.0)[A] | 1.9 (1.0)* |
| **Race/ethnicity[b]** | Mexican American | 56.8 (0.6)[A] | 112.9 (1.1)[B] | 47.7 (1.1)[AB] | 6.6 (1.0)[A] | 7.7 (1.0)[A] | 20.8 (1.1)[AB] | 6.4 (1.0)[A] | 1.7 (1.1)[A] | 2.3 (1.1)[B] |
| | Other Hispanic | 53.5 (0.9)[B] | 101.0 (1.0)[AB] | 58.9 (1.1)[B] | 7.0 (1.0)[A] | 8.9 (1.1)[AB] | 18.9 (1.1)[AB] | 6.4 (1.1)[A] | 1.7 (1.1)[A] | 2.3 (1.1)[B] |
| | Non-Hispanic White | 59.6 (0.5)[C] | 94.2 (1.0)[A] | 57.2 (1.1)[B] | 8.1 (1.0)[B] | 9.7 (1.0)[B] | 19.8 (1.0)[AB] | 6.5 (1.0)[A] | 2.5 (1.1)[B] | 1.6 (1.0)[A] |
| | Non-Hispanic Black | 61.4 (0.8)[C] | 84.8 (1.0)[C] | 41.6 (1.1)[A] | 6.4 (1.0)[A] | 7.1 (1.1)[A] | 21.5 (1.0)[B] | 6.4 (1.0)[A] | 2.1 (1.1)[AB] | 2.0 (1.0)[B] |
| | Other Race (including Multi-Racial) | 48.6 (1.0)[D] | 99.4 (1.0)[AB] | 43.8 (1.1)[A] | 6.1 (1.0)[A] | 7.4 (1.1)[A] | 17.0 (1.1)[A] | 5.9 (1.0)[A] | 2.0 (1.1)[AB] | 1.8 (1.1)[AB] |
| **Income to poverty[b]** | 0.00–1.30 | 60.5 (0.7)[C] | 103.5 (1.0)[B] | 49.4 (1.0)[A] | 6.9 (1.0)[A] | 8.9 (1.0)[AB] | 26.9 (1.0)[A] | 7.0 (1.0)[B] | 2.0 (1.0)[A] | 2.0 (1.1)[B] |
| | >1.30–3.50 | 59.5 (0.7)[BC] | 92.7 (1.0)[A] | 50.0 (1.0)[A] | 7.4 (1.0)[B] | 8.7 (1.0)[AB] | 21.4 (1.0)[B] | 6.7 (1.0)[B] | 2.4 (1.1)[A] | 1.8 (1.1)[AB] |
| | >3.50 and above | 56.3 (0.6)[A] | 92.7 (1.0)[A] | 60.3 (1.1)[B] | 8.1 (1.0)[C] | 9.3 (1.1)[B] | 15.3 (1.0)[C] | 6.0 (1.0)[A] | 2.4 (1.1)[A] | 1.6 (1.1)[A] |
| | missing | 56.2 (1.2)[AB] | 98.9 (1.1)[AB] | 45.0 (1.1)[A] | 7.1 (1.1)[ABC] | 7.6 (1.1)[A] | 19.6 (1.1)[B] | 6.2 (1.1)[AB] | 1.9 (1.1)[A] | 2.1 (1.1)[B] |
| **Cycle** | 2009–10 | 58.3 (0.8) | 156.1 (1.1) | 47.5 (1.1) | 8.9 (1.0) | 12.7 (1.1) | 26.3 (1.1) | 8.5 (1.0) | _ | _ |
| | 2011–12 | 60.1 (0.9) | 107.7 (1.0) | 77.3 (1.1) | 8.4 (1.0) | 13.5 (1.1) | 20.0 (1.0) | 7.5 (1.0) | _ | _ |
| | 2013–14 | 58.5 (0.8) | 77.8 (1.0) | 73.1 (1.0) | 8.3 (1.0) | 8.5 (1.1) | 17.4 (1.0) | 5.7 (1.0) | 2.7 (1.1) | 1.7 (1.1) |
| | 2015–16 | 56.4 (0.7)* | 65.4 (1.0)* | 30.1 (1.1)* | 5.2 (1.0)* | 4.5 (1.1)* | 17.0 (1.1)* | 4.8 (1.0)* | 1.9 (1.0)[£] | 1.9 (1.1) |
| **BMI[b]** | underweight | 57.0 (1.5)[AB] | 92.7 (1.1) | 53.9 (1.1)[A] | 7.6 (1.1)[A] | 9.0 (1.1) | 20.4 (1.1) | 7.2 (1.1)[A] | 3.0 (1.2)[A] | 1.8 (1.1)[A] |
| | normoweight | 59.3 (0.6)[B] | 102.9 (1.0) | 53.8 (1.0)[A] | 7.8 (1.0)[A] | 9.6 (1.0) | 22.0 (1.0) | 6.6 (1.0)[A] | 2.1 (1.1)[A] | 1.7 (1.1)[A] |
| | overweight | 56.3 (0.6)[A] | 91.1 (1.0) | 52.1 (1.1)[A] | 7.3 (1.0)[A] | 8.6 (1.0) | 17.1 (1.0) | 6.3 (1.0)[A] | 2.3 (1.1)[A] | 1.7 (1.1)[A] |
| | obesity | 59.2 (0.5)[B] | 92.9 (1.0)* | 53.1 (1.0)[A] | 7.4 (1.0)[A] | 8.5 (1.0)* | 20.2 (1.0)* | 6.4 (1.0)[A] | 2.3 (1.1)[A] | 1.9 (1.1)[A] |
| **Physical activity[b]** | Low | 58.5 (0.6) | 92.3 (1.0)[A] | 49.7 (1.0)[A] | 6.8 (1.0) | 8.1 (1.0) | 19.2 (1.0)[A] | 6.2 (1.0)[A] | 2.3 (1.1)[A] | 1.8 (1.1)[A] |
| | Medium | 58.3 (0.8) | 102.7 (1.0)[B] | 58.2 (1.1)[B] | 8.2 (1.0) | 9.8 (1.1) | 19.9 (1.0)[A] | 6.6 (1.0)[A] | 2.2 (1.1)[A] | 1.8 (1.1)[A] |
| | High | 58.2 (0.5) | 95.5 (1.0)[AB] | 53.8 (1.0)[AB] | 7.8 (1.0)* | 9.2 (1.0)* | 20.2 (1.0)[A] | 6.6 (1.0)[A] | 2.3 (1.1)[A] | 1.8 (1.0)[A] |
| **Current smoker** | no | 58.0 (0.4) | 96.8 (1.0) | 54.9 (1.0) | 7.8 (1.0) | 9.1 (1.0) | 19.4 (1.0) | 6.3 (1.0) | 2.2 (1.0) | 1.7 (1.0) |
| | yes | 59.7 (0.9)[£] | 89.3 (1.0)[£] | 44.5 (1.1)[£] | 6.4 (1.0)[£] | 8.1 (1.0)[£] | 22.1 (1.0)[£] | 6.9 (1.0)[£] | 2.8 (1.1)[£] | 2.0 (1.1)[£] |
| **Energy above recommended** | no | 57.1 (0.5) | 92.4 (1.0) | 52.2 (1.0) | 7.3 (1.0) | 8.7 (1.0) | 19.5 (1.0) | 6.3 (1.0) | 2.2 (1.1) | 1.7 (1.0) |
| | yes | 60.3 (0.5)[£] | 100.6 (1.0)[£] | 54.4 (1.0) | 7.8 (1.0)[£] | 9.2 (1.0)[£] | 20.3 (1.0) | 6.6 (1.0) | 2.3 (1.1) | 1.9 (1.0) |
| **Total fat intake (% of total energy intake) above >30%** | no | 54.6 (0.6) | 97.8 (1.0) | 49.5 (1.0) | 7.1 (1.0) | 8.7 (1.0) | 20.2 (1.0) | 6.7 (1.0) | 2.1 (1.1) | 1.8 (1.1) |
| | yes | 60.1 (0.4)[£] | 94.4 (1.0) | 54.9 (1.0)[£] | 7.7 (1.0)[£] | 9.0 (1.0) | 19.6 (1.0) | 6.3 (1.0)[£] | 2.3 (1.0) | 1.8 (1.0) |

*(Continued)*

**Table 1.** (Continued)

| | | Dietary contribution of ultra-processed foods (% total energy intake) (n = 9,416) | PHTHALATES (ηmol/g creatinine) | | | | | BISPHENOL (ηmol/g creatinine)[c] | | |
|---|---|---|---|---|---|---|---|---|---|---|
| | | | ΣDEHP (n = 9,416) | ΣDiNP (n = 9,416) | mCNP (n = 9,416) | mCPP (n = 9,416) | mBzP (n = 9,416) | BPA (n = 9,420) | BPF (n = 4,655) | BPS (n = 4,655) |
| | | mean (SE) | GM[a] (GSE) | GM (GSE) | GM (GSE) | GM (GSE) | GM (GSE) | GM (GSE) | GM (GSE) | GM (GSE) |
| Fat in UPF derived total energy intake (% of total energy intake) above median (18.7%) | no | 44.7 (0.4) | 94.2 (1.0) | 47.5 (1.0) | 7.0 (1.0) | 8.2 (1.0) | 18.9 (1.0) | 6.2 (1.0) | 2.0 (1.0) | 1.9 (1.0) |
| | yes | 71.9 (0.3)[£] | 96.8 (1.0) | 59.3 (1.0)[£] | 8.0 (1.0)[£] | 9.6 (1.0)[£] | 20.7 (1.0)[£] | 6.6 (1.0)[£] | 2.5 (1.1)[£] | 1.7 (1.0)[£] |
| **Total** | | 58.3 (0.4) | 95.5 (1.0) | 53.1 (1.0) | 7.5 (1.0) | 8.9 (1.0) | 19.8 (1.0) | 6.4 (1.0) | 2.3 (1.0) | 1.8 (1.0) |

[a]GM = Geometric means; GSE = Geometric standard error

[b]Values sharing a letter in the group label are not significantly different at the p<0.05 level (using Bonferroni inequality adjustment for multiple comparisons).

[c]Bisphenol F and S analysis were only measured in 2013–2016 cycles

[*]Statistically significant linear trend (p<0.05)

[£]Statistically significant (p<0.05)

to 23.3%) higher levels of mBzP, 6.2% (95% CI: -2.7% to 15.9%) higher levels of BPA, and 33.8% (95% CI: 11.7% to 60.3%) higher levels of BPF. On the other hand, the highest quintile of ultra-processed food had 25.1% (95% CI: 12.0 to 36.2%) lower levels of BPS. The tendencies remained stable when ultra-processed food was expressed as % of total gram intake instead of % of total energy though the association with mCNP, mCPP and BPS lost statistical significance and the association with BPA gained significance (S2 Table).

Dose-response curve between dietary contribution of ultra-processed food and urinary biomarker concentrations using restricted cubic splines are displayed in S1 Fig. There was evidence of a dose-response association with no departure from linearity (p>0.05 for linearity) for mCPP and ΣDiNP.

In sensitivity analyses, additional adjustment for total fat intake did not change the main effects, though the positive association with mCPP lost statistical significance (S3 Table). When adjusting for fast food intake, the association with ΣDiNP and mCPP became non-significant. Further adjustment for creatinine concentration (as covariate), did not change the main effects though the association with BPA became significant. The strength of the association with urinary concentrations remained virtually the same when using quintiles of fat in ultra-processed food derived total energy intake (% of total energy intake) except for mBzP which became non-significant (S4 Table).

The association between ultra-processed food and urinary concentrations were not modified by cycle, age or sex, except ΣDiNP and BPA. The association of ultra-processed food intake with ΣDiNP was stronger in children than in adults and did not reach statistical significance among adolescents (p for interaction = 0.08). A positive non-significant association between ultra-processed food and BPA was found in both men and women, though the association was stronger in men (p for interaction = 0.02) (S5 Table).

Though exposure to phthalates and bisphenol has declined since 2009 (except for BPS), the trends of association between quintiles of UPF and phthalates and bisphenol concentrations in each cycle mirror the associations observed across all cycles (S6 Table).

We observed a monotonic increase of ΣDiNP, mCNP, mCPP, mBzP (p for trend ≤ 0.001) and BPA (p for trend = 0.042) with tertiles of Ready-to-heat ultra-processed foods. Non-

**Table 2. Phthalate/Bisphenol levels according to the quintiles of the dietary contribution of ultra-processed foods.** Subsample of US population aged 6 + years (NHANES 2009–2016).

| | | Quintile of dietary contribution of ultra-processed foods (% of total energy intake)[a] | | | | | |
|---|---|---|---|---|---|---|---|
| | | Q1 | Q2 | Q3 | Q4 | Q5 | p for trend |
| ΣDEHP (GM[b]) | Crude (ηmol/mL) | 0.08 | 0.09 | 0.09 | 0.09 | 0.10 | <0.001 |
| | Standardized (ηmol/g creatinine) | 93.3 | 96.2 | 91.9 | 96.5 | 100.0 | 0.075 |
| | Adjusted for socio-demographic variables (ηmol/g creat)[c] | 97.9 | 96.5 | 91.7 | 95.5 | 96.0 | 0.493 |
| | Adjusted for socio-demographic + other variables (ηmol/g creat)[d] | 98.4 | 96.4 | 91.5 | 95.3 | 96.0 | 0.425 |
| ΣDiNP (GM) | Crude (ηmol/mL) | 0.04 | 0.05 | 0.05 | 0.06 | 0.06 | <0.001 |
| | Standardized (ηmol/g creatinine) | 44.5 | 52.1 | 51.3 | 58.0 | 61.0 | <0.001 |
| | Adjusted for socio-demographic variables (ηmol/g creat)[c] | 47.0 | 52.5 | 50.7 | 58.3 | 57.7 | 0.001 |
| | Adjusted for socio-demographic + other variables (ηmol/g creat)[d] | 46.9 | 52.4 | 50.6 | 58.2 | 57.9 | 0.001 |
| mCNP (GM) | Crude (ηmol/mL) | 0.006 | 0.007 | 0.007 | 0.008 | 0.008 | <0.001 |
| | Standardized (ηmol/g creatinine) | 6.5 | 7.4 | 7.4 | 8.1 | 8.1 | <0.001 |
| | Adjusted for socio-demographic variables (ηmol/g creat)[c] | 6.9 | 7.5 | 7.3 | 8.1 | 7.8 | 0.001 |
| | Adjusted for socio-demographic + other variables (ηmol/g creat)[d] | 6.9 | 7.4 | 7.3 | 8.1 | 7.9 | 0.001 |
| mCPP (GM) | Crude (ηmol/mL) | 0.007 | 0.008 | 0.008 | 0.009 | 0.011 | <0.001 |
| | Standardized (ηmol/g creatinine) | 7.8 | 8.9 | 8.2 | 9.7 | 10.1 | <0.001 |
| | Adjusted for socio-demographic variables (ηmol/g creat)[c] | 8.4 | 9.1 | 8.1 | 9.5 | 9.4 | 0.039 |
| | Adjusted for socio-demographic + other variables (ηmol/g creat)[d] | 8.4 | 9.1 | 8.1 | 9.5 | 9.4 | 0.035 |
| mBzP (GM) | Crude (ηmol/mL) | 0.02 | 0.02 | 0.02 | 0.02 | 0.03 | <0.001 |
| | Standardized (ηmol/g creatinine) | 17.3 | 17.8 | 19.1 | 21.6 | 24.1 | <0.001 |
| | Adjusted for socio-demographic variables (ηmol/g creat)[c] | 19.1 | 18.7 | 19.3 | 20.6 | 21.5 | 0.007 |
| | Adjusted for socio-demographic + other variables (ηmol/g creat)[d] | 19.2 | 18.8 | 19.3 | 20.5 | 21.3 | 0.017 |
| BPA (GM) | Crude (ηmol/mL) | 0.005 | 0.006 | 0.006 | 0.007 | 0.007 | <0.001 |
| | Standardized (ηmol/g creatinine) | 6.1 | 6.1 | 6.3 | 6.7 | 6.9 | 0.001 |
| | Adjusted for socio-demographic variables (ηmol/g creat)[c] | 6.3 | 6.2 | 6.3 | 6.7 | 6.7 | 0.046 |
| | Adjusted for socio-demographic + other variables (ηmol/g creat)[d] | 6.3 | 6.2 | 6.3 | 6.7 | 6.7 | 0.056 |
| BPF (GM) | Crude (ηmol/mL) | 0.002 | 0.002 | 0.002 | 0.003 | 0.003 | <0.001 |
| | Standardized (ηmol/g creatinine) | 1.8 | 2.3 | 2.3 | 2.5 | 2.4 | 0.001 |
| | Adjusted for socio-demographic variables (ηmol/g creat)[c] | 1.8 | 2.3 | 2.3 | 2.6 | 2.4 | 0.003 |
| | Adjusted for socio-demographic + other variables (ηmol/g creat)[d] | 1.8 | 2.3 | 2.3 | 2.6 | 2.4 | 0.004 |
| BPS (GM) | Crude (ηmol/mL) | 0.0020 | 0.0017 | 0.0017 | 0.0017 | 0.0018 | 0.108 |
| | Standardized (ηmol/g creatinine) | 2.2 | 1.8 | 1.8 | 1.7 | 1.6 | <0.001 |
| | Adjusted for socio-demographic variables (ηmol/g creat)[c] | 2.1 | 1.8 | 1.8 | 1.7 | 1.6 | 0.002 |
| | Adjusted for socio-demographic + other variables (ηmol/g creat)[d] | 2.1 | 1.8 | 1.8 | 1.7 | 1.6 | 0.001 |

[a]For all phthalates: Mean (range) dietary contribution of ultra-processed foods per quintile: 1st = 27.1 (0 to 39.5); 2nd = 46.8 (39.5 to 53.3); 3rd = 59.3 (53.3 to 65.2); 4th = 71.1 (65.2 to 77.7); 5th = 87.3 (77.7 to 100)

[b]Geometric means (GM) presented in all cases

[c]Adjusted for cycle, sex, age group (6 to 11, 12 to 19, +20), race/ethnicity (Mexican American, Other Hispanic, Non-Hispanic White, Non-Hispanic Black, Other Race), ratio of family income to poverty (Supplemental Nutrition Assistance Program 0.00–1.30, >1.30–3.50 and >3.50 and over).

[d]Adjusted for cycle, sex, age group, race/ethnicity, and income, in addition to energy intake above recommended levels (y/n), BMI (underweight, normoweight, overweight, obesity), physical activity (low, medium, high) and current smoking (y/n).

Ready-to-heat ultra-processed foods were positively associated with BPF concentration (p for trend = 0.005) and inversely associated with BPS (p for trend ≤ 0.001) (Table 3).

An inverse association was observed between quintiles of the dietary contribution of minimally processed foods and ΣDiNP, mCNP, mCPP, mBzP, BPA and BPF, and a positive

**Table 3. Phthalate/Bisphenol levels according to tertiles of dietary contribution of ultra-processed ready-to-heat and all remaining subgroups[a].** Subsample of US population aged 6 + years (NHANES 2009–2016).

| | | | PHTHALATES (ηmol/g creatinine) | | | | | BISPHENOL (ηmol/g creatinine) | | |
|---|---|---|---|---|---|---|---|---|---|---|
| | | | ΣDEHP | ΣDiNP | mCNP | mCPP | mBzP | BPA | BPF | BPS |
| Tertiles of dietary contribution of ultra-processed food subgroups (% of total energy intake) | Ready-to-heat food subgroup[a] | T1 | 94.6 | 47.1 | 7.1 | 8.4 | 19.2 | 6.3 | 2.1 | 1.9 |
| | | T2 | 99.6 | 58.0 | 7.4 | 9.0 | 20.3 | 6.4 | 2.7 | 1.6 |
| | | T3 | 95.5 | 62.4 | 8.2 | 9.8 | 20.6 | 6.7 | 2.3 | 1.7 |
| | | p for trend | 0.644 | <0.001 | <0.001 | <0.001 | <0.001 | 0.042 | 0.164 | 0.056 |
| | Remaining subgroups[b] | T1 | 97.6 | 53.6 | 7.4 | 9.0 | 19.9 | 6.3 | 1.9 | 2.0 |
| | | T2 | 94.6 | 53.4 | 7.7 | 9.0 | 19.1 | 6.6 | 2.5 | 1.7 |
| | | T3 | 94.4 | 52.2 | 7.5 | 8.7 | 20.4 | 6.4 | 2.4 | 1.6 |
| | | p for trend | 0.208 | 0.573 | 0.586 | 0.364 | 0.487 | 0.525 | 0.005 | <0.001 |

All models adjusted for sex, age group (6 to 11, 12 to 19, +20), race/ethnicity (Mexican American, Other Hispanic, Non-Hispanic White, Non-Hispanic Black, Other Race), ratio of family income to poverty (Supplemental Nutrition Assistance Program 0.00–1.30, >1.30–3.50 and >3.50 and over)

[a]For all phthalates: Mean (range) dietary contribution of ready-to-heat ultra-processed foods per tertile: 1st = 0 (0 to 0); 2nd = 6.5 (0.1 to 11.4); 3rd = 30.7 (11.4 to 100)

[b]For all phthalates: Mean (range) dietary contribution of remaining ultra-processed foods per tertile: 1st = 25.9 (0 to 38.7); 2nd = 47.3 (38.7 to 55.8); 3rd = 68.9 (55.8 to 100)

association with BPS (S7 Table). Quintiles of the dietary contribution of processed culinary ingredients were positively associated with ΣDEHP and BPS and inversely associated with ΣDiNP and BPF (S8 Table). A lack of association was observed between quintiles of the dietary contribution of processed foods and phthalate or bisphenol levels (S9 Table).

## Discussion

In this cross-sectional study of the US population aged 6 + years, there was evidence of monotonic dose-response association between ultra-processed food consumption (expressed as % of total energy intake) and urinary concentration of ΣDiNP, mCNP, mCPP, mBzP and BPF. No association was observed with ΣDEHP, BPA and an inverse association was observed with BPS. The association with BPA gained significance when ultra-processed food was expressed as % of total gram intake. These associations were largely consistent across age and sex subpopulations and remained significant after adjusting for total fat intake. Though exposures to phthalates and bisphenol have declined since 2009 (except for BPS) probably reflecting the effect of legislative activity and the advocacy efforts of nongovernmental organizations on the use of phthalates in consumer products and consumer behavior [43], these associations were also consistent across cycles. A previous study restricted to data from NHANES 2013–14 also reported a positive association between ultra-processed food consumption and mCNP, mCPP and Mono-(carboxyisoctyl) phthalate and a lack of association with ΣDEHP and BPA, however it did not observe a significant positive association with MBzP or BPF or an inverse association with BPS [30].

The lack of association of ΣDEHP with ultra-processed food quintiles was also observed with quintiles of minimally processed foods and processed foods. However, we did find an association with quintiles of processed culinary ingredients, which may be explained by the lipophilic nature of ΣDEHP which tend to concentrate in fattier foods such as butter, cream, cooking oils and animal fats (processed culinary ingredients) [60, 65].

While a positive though non-significant association was observed between BPA and both ultra-processed foods and processed foods, we found an inverse association with quintiles of minimally processed foods, and a lack of association with processed culinary ingredients. These findings may be explained by the fact that canned food which are mainly processed and ultra-processed foods, are considered the predominant source of BPA [66]. Indeed, contamination of food with BPA is usually caused by contact with food packaging materials containing epoxy resins and polycarbonate. Epoxy resins are often used as internal coatings of cans to protect from rusting and corrosion and to prevent direct contact of food with metal can walls, and in metal lids for in glass food jars. BPA in polycarbonate containers and coatings can migrate into foods, during storage and processing at elevated temperatures [66, 67].

Due to concerns regarding the health effects of BPA, industries have sought for alternatives such as BPF and BPS [68]. In this study we observed a positive association between ultra-processed food consumption and BPF levels but a negative association with BPS concentration. In recent years, BPS has been used as a substitute for BPA in thermal papers, while high levels of BPS and low levels of BPA have been found in thermal register receipts. What is still unknown, however, is whether BPS has been used as a substitute for BPA in can coatings [68]. In at least one study, BPS was not detected in any of the canned food composite samples and was detected instead in samples prepared from meat and meat products, indicating that sources of BPS other than can coatings are possible [68].

Epidemiological evidence on food sources of ΣDiNP, mCNP and mBzP is scarce [60, 69]. In our study, ΣDiNP, mCNP and mBzP were positively associated with ultra-processed food consumption. Some studies have suggested that fast food consumption may be a unique source of ΣDiNP [62, 63] and mBzP exposure [63]. Interestingly, when we further adjusted for dietary contribution of fast food, the association between ultra-processed food and ΣDiNP lost statistical significance but not the association with mBzP.

There is some epidemiological evidence of association between consumption of meats and fatty foods such as dairy and MCPP levels [60]. Consistent with these results, we observed a positive association between ultra-processed food consumption and MCPP concentrations which became non-significant with further adjustment by total fat intake or fast food intake.

Our study further suggests that ΣDiNP, mCNP, mCPP, mBzP and BPA may be more concentrated in ready-to-eat ultra-processed foods often heated or served in paper, plastic or cardboard containers, while non-ready-to-heat ultra-processed foods were directly associated with BPF and inversely associated with BPS. Though little is known about the migration of phthalates and bisphenol from food packaging during heating, at least one study observed a correlation between migration of dibutyl phthalate (DBP) and heating time [29]. Another study concluded that paper and cardboard used in food packaging may contribute to the inadvertent exposure of consumers to endocrine-disrupting chemicals [28].

We observed that the association of ultra-processed food intake with ΣDiNP was stronger in children than in adults and did not reach statistical significance among adolescents. Similarly, Buckley reported a stronger association between ultra-processed food intake and MCOP among children as compared with adults or adolescents [30]. Differences in types of ultra-processed foods consumed or metabolism between age groups may explain these results.

In our study we observed a positive association between ultra-processed food and urinary concentration of most phthalates and bisphenol, suggesting that contamination by contact materials may be an additional pathway to explain the associations seen between ultra-processed food and various health outcomes, as previously suggested by Srour [14]. Indeed, BPA exposure has been linked with higher risk of diabetes, general/abdominal obesity and hypertension [25] and phthalates with diabetes [26] and insulin resistance [27].

There are several strengths to this study including the use of a large, nationally representative sample of the US population, increasing the external validity of results. The disaggregation of recipes into underlying ingredients enabled the calculation of more precise estimates of dietary contribution of ultra-processed foods. The use of dietary contribution of ultra-processed food as exposure should reduce bias introduced by non-differential calorie misreporting from all foods.

Some study limitations must be acknowledged. The observational nature and cross-sectional design of NHANES does not allow the inference of causal relationship between ultra-processed food consumption and urinary phthalate or bisphenol concentrations. However, given the short biologic half-lives (<24 hour) of both phthalates and bisphenol [39], both urine samples and dietary information represent exposures during approximately the same 24-hour period. While multiple or 24-h urine samples are the ideal, reliance on a single spot urine sample corresponds well with short elimination half-lives [40].

Though self-reported dietary data are liable to information bias, 24-hour recalls as used in this study have been shown to be the least-biased self-report instrument available [70]. Additionally, standardized methods for assessing population intakes in NHANES have been shown to produce accurate intake estimates [34–36]. Differential underreporting of ultra-processed food consumption driven by social desirability bias could lead to underestimation of ultra-processed food dietary contribution or dilute the association between ultra-processed food consumption and urinary concentrations. Though NHANES collects limited information indicative of food processing (i.e. place of meals or product brand names), this is not consistently provided for all food items, which could lead to modest under or overestimation of the dietary contribution of ultra-processed foods. Lastly, the observed associations may be influenced by residual confounders such as source of food (i.e. food away from home, fast food or vending machine), exposure to materials in contact with foodstuffs, oral contact materials other than food (i.e. toys), dermal contact, dust in the environment or occupational exposure [66].

In conclusion, our findings suggest that ultra-processed food consumption may be a source of exposure to ΣDiNP, mCNP, mCPP, mBzP and BPF in the US population. Future studies should seek to confirm our findings and extend the research to examine health outcomes. As both phthalates/bisphenol and ultra-processed food have been previously linked with insulin resistance, diabetes, general/abdominal obesity and hypertension, future longitudinal studies may help to better understand the mediating role of contact materials in the association between ultra-processed food consumption and these outcomes.

## Supporting information

**S1 Fig. Phthalate/Bisphenol levels standardized in ηmol/g creatinine regressed on the dietary contribution of ultra-processed foods (% of total energy intake) using restricted cubic splines.** The values shown on the x-axis correspond to the 5th, 27.5th, 50th, 72.5th, and 95th centiles for percentage of total energy from ultra-processed foods (knots). All regressions were adjusted for cycle, sex, age group (6 to 11, 12 to 19, +20), race/ethnicity (Mexican American, Other Hispanic, Non-Hispanic White, Non-Hispanic Black, Other Race), ratio of family income to poverty (Supplemental Nutrition Assistance Program 0.00–1.30, >1.30–3.50 and >3.50 and over), energy intake above recommended levels (y/n), BMI (underweight, normoweight, overweight, obesity), physical activity (low, medium, high) and current smoking (y/n).

- Di(2-ethylhexyl) phthalate (nmol/g creatinine): Wald test for linear term p = 0.354; Wald test for all non-linear terms p = 0.4203).

- Di-isononyl phthalate (nmol/g creatinine): There was little evidence of non-linearity in the restricted cubic spline model (Wald test for linear term p = 0.005; Wald test for all non-linear terms p = 0.3945).

- Mono(carboxynonyl) Phthalate (nmol/g creatinine): There was evidence of non-linearity in the restricted cubic spline model (Wald test for linear term p = 0.031; Wald test for all non-linear terms p = 0.4646).

- Mono(3-carboxypropyl) phthalate (nmol/g creatinine): Wald test for linear term p = 0.056; Wald test for all non-linear terms p = 0.4721.

- Mono-benzyl phthalate (nmol/g creatinine): Wald test for linear term p = 0.276; Wald test for all non-linear terms p = 0.1786.

- BPA: Wald test for linear term p = 0.177; Wald test for all non-linear terms p = 0.3955.

- BPF: Wald test for linear term p = 0.493; Wald test for all non-linear terms p = 0.2131.

- BPS: Wald test for linear term p = 0.318; Wald test for all non-linear terms p = 0.7867.
(XLSX)

**S1 Table. Summary of phthalate parent compounds/metabolites, bisphenols, and their proposed sources (based on Pacyga et al. 2019).**
(XLSX)

**S2 Table. Phthalate/Bisphenol levels according to the quintiles of dietary contribution of ultra-processed foods (% of total gram intake).** Subsample of US population aged 6 + years (NHANES 2009–2016). Legend. aFor all phthalates: Mean (range) dietary contribution of processed foods (% of total gram intake) per quintile [n]: 1st = 7.2 (0 to 12.4)%; 2nd = 17.2 (12.4 to 22.1)%; 3rd = 27.8 (22.1 to 34.2)%; 4th = 42.0 (34.2 to 51.3)%; 5th = 68.2 (51.3 to 100)%. All analysis adjusted for cycle, sex, age group (6 to 11, 12 to 19, +20), race/ethnicity (Mexican American, Other Hispanic, Non-Hispanic White, Non-Hispanic Black, Other Race), ratio of family income to poverty (Supplemental Nutrition Assistance Program 0.00–1.30, >1.30–3.50 and >3.50 and over) + energy intake above recommended levels (y/n), BMI (underweight, normoweight, overweight, obesity), physical activity (low, medium, high) and current smoking (y/n). Geometric mean (GM) presented in all cases.
(XLSX)

**S3 Table. Sensitivity analysis of the association between dietary contribution of ultra-processed foods and phthalate/bisphenol levelsa.** Subsample of US population aged 6 + years (NHANES 2009–2016). Legend: aAll analysis adjusted for cycle, sex, age group (6 to 11, 12 to 19, +20), race/ethnicity (Mexican American, Other Hispanic, Non-Hispanic White, Non-Hispanic Black, Other Race), ratio of family income to poverty (Supplemental Nutrition Assistance Program 0.00–1.30, >1.30–3.50 and >3.50 and over), energy intake above recommended levels (y/n), BMI (underweight, normoweight, overweight, obesity), physical activity (low, medium, high) and current smoking (y/n). bGM = geometric mean. cAditionally adjusted for total fat intake (% of total energy intake). dAditionally adjusted for fast food intake (% of total energy intake). eAditionally adjusted for creatinine concentration (milligrams/dL) (Ln).
(XLSX)

**S4 Table. Phthalate/Bisphenol levels standardized in ηmol/g creatinine according to the quintiles of dietary contribution of total fat in UPF derived energy intake (% of total energy intake).** Subsample of US population aged 6 + years (NHANES 2009–2016) (n = 9,416 for phthalates and n = 9,320 for bisphenol). Legend: aMean (range) dietary contribution of

UPF derived total fat intake per quintile: 1st = 6.3 (0 to 10.2); 2nd = 13.3 (10.2 to 16.1); 3rd = 18.8 (16.1 to 21.6); 4th = 24.6 (21.6 to 28.1); 5th = 34.1 (28.1 to 58.6). sAll analysis adjusted for cycle, sex, age group (6 to 11, 12 to 19, +20), race/ethnicity (Mexican American, Other Hispanic, Non-Hispanic White, Non-Hispanic Black, Other Race), ratio of family income to poverty (Supplemental Nutrition Assistance Program 0.00–1.30, >1.30–3.50 and >3.50 and over) + energy intake above recommended levels (y/n), BMI (underweight, normo-weight, overweight, obesity), physical activity (low, medium, high) and current smoking (y/n). Geometric mean (GM) presented in all cases.
(XLSX)

**S5 Table. Phthalate/Bisphenol levels standardized in ηmol/g creatinine according to the quintiles of the dietary contribution of ultra-processed foods (% of total energy intake), stratified by covariates with statistically significant interaction.** Subsample of US population aged 6 + years (NHANES 2009–2016). Legend: aGM = geometric mean. All analysis were adjusted for cycle, sex, age group (6 to 11, 12 to 19, +20), race/ethnicity (Mexican American, Other Hispanic, Non-Hispanic White, Non-Hispanic Black, Other Race), ratio of family income to poverty (Supplemental Nutrition Assistance Program 0.00–1.30, >1.30–3.50 and >3.50 and over), education (<12y, 12y, >12y), energy intake above recommended levels (y/ n), BMI (underweight, normoweight, overweight, obesity), physical activity (low, medium, high) and current smoking (y/n).
(XLSX)

**S6 Table. Phthalate/Bisphenol levels (standardized in ηmol/g creatinine) according to the quintiles of the dietary contribution of ultra-processed foods (% of total energy intake) in each cycle.** Subsample of US population aged 6 + years (NHANES 2009–2016). Legend: aMean (range) dietary contribution of ultra-processed foods per quintile: 2009–10: 1st = 27.9 (0 to 40.2); 2nd = 47.0 (40.3 to 53.3); 3rd = 59.2 (53.3 to 64.7); 4th = 70.9 (64.8 to 77.1); 5th = 86.3 (77.1 to 100). 2011–12: 1st = 28.7 (0 to 40.8); 2nd = 48.6 (40.8 to 55.6); 3rd = 61.5 (55.7 to 67.2); 4th = 73.0 (67.2 to 79.7); 5th = 89.0 (79.7 to 100). 2013–14: 1st = 27.4 (0 to 40.1); 2nd = 47.2 (40.1 to 54.1); 3rd = 59.1 (54.2 to 64.8); 4th = 71.3 (64.8 to 78.5); 5th = 87.8 (78.5 to 100). 2015–16: 1st = 24.8 (0 to 36.8); 2nd = 44.6 (36.8 to 51.6); 3rd = 57.4 (51.6 to 63.6); 4th = 69.5 (63.7 to 75.6); 5th = 86.0 (75.6 to 100). bGM = geometric mean. All analysis adjusted for cycle, sex, age group, race/ethnicity, and income, in addition to energy intake above recommended levels (y/n), BMI (underweight, normoweight, overweight, obesity), physical activity (low, medium, high) and current smoking (y/n).
(XLSX)

**S7 Table. Phthalate/Bisphenol levels according to the quintiles of dietary contribution of minimally processed foods (% of total energy intake).** Subsample of US population aged 6 + years (NHANES 2009–2016). Legend: aFor all phthalates: Mean (range) dietary contribution of unprocessed/ minimally processed foods per quintile [n]: 1st = 5.8 (0 to 11.9)% [n = 1784]; 2nd = 16.7 (15.8 to 23.6)% [n = 1746]; 3rd = 25.7 (23.6 to 31.3) [n = 1773]; 4th = 36.2 (31.3 to 41.3) [n = 1928]; 5th = 55.4 (41.3 to 100) [n = 2185]. All analysis adjusted for cycle, sex, age group (6 to 11, 12 to 19, +20), race/ethnicity (Mexican American, Other Hispanic, Non-His-panic White, Non-Hispanic Black, Other Race), ratio of family income to poverty (Supplemen-tal Nutrition Assistance Program 0.00–1.30, >1.30–3.50 and >3.50 and over) + energy intake above recommended levels (y/n), BMI (underweight, normoweight, overweight, obesity), physical activity (low, medium, high) and current smoking (y/n). Geometric mean (GM) pre-sented in all cases.
(XLSX)

**S8 Table. Phthalate/Bisphenol levels according to the quintiles of dietary contribution of processed culinary ingredients (% of total energy intake).** Subsample of US population aged 6 + years (NHANES 2009–2016). Legend: aFor all phthalates: Mean (range) dietary contribution of processed cunilary ingredients per quintile [n]: 1st = 0 (0 to 0)% [n = 2701]; 2nd = 0.8 (0.003 to 1.4)% [n = 1071]; 3rd = 2.5 (1.4 to 3.6)% [n = 1782]; 4th = 5.3 (3.6 to 7.3)% [n = 1891]; 5th = 12.5 (7.3 to 51.4)% [n = 1971]. All analysis adjusted for cycle, sex, age group (6 to 11, 12 to 19, +20), race/ethnicity (Mexican American, Other Hispanic, Non-Hispanic White, Non-Hispanic Black, Other Race), ratio of family income to poverty (Supplemental Nutrition Assistance Program 0.00–1.30, >1.30–3.50 and >3.50 and over) + energy intake above recommended levels (y/n), BMI (underweight, normoweight, overweight, obesity), physical activity (low, medium, high) and current smoking (y/n). Geometric mean (GM) presented in all cases.
(XLSX)

**S9 Table. Phthalate/Bisphenol levels according to the quintiles of dietary contribution of processed foods (% of total energy intake).** Subsample of US population aged 6 + years (NHANES 2009–2016). Legend: aFor all phthalates: Mean (range) dietary contribution of processed foods per quintile [n]: 1st = 0 (0 to 0)% [n = 2280]; 2nd = 1.5 (0.003 to 3.6)% [n = 2038]; 3rd = 5.9 (3.6 to 8.6)% [n = 1915]; 4th = 12.2 (8.6 to 16.7)% [n = 1713]; 5th = 28.2 (16.7 to 88.3)% [n = 1470]. All analysis adjusted for cycle, sex, age group (6 to 11, 12 to 19, +20), race/ethnicity (Mexican American, Other Hispanic, Non-Hispanic White, Non-Hispanic Black, Other Race), ratio of family income to poverty (Supplemental Nutrition Assistance Program 0.00–1.30, >1.30–3.50 and >3.50 and over) + energy intake above recommended levels (y/n), BMI (underweight, normoweight, overweight, obesity), physical activity (low, medium, high) and current smoking (y/n). Geometric mean (GM) presented in all cases.
(XLSX)

## Author Contributions

**Conceptualization:** Eurídice Martínez Steele, Neha Khandpur, Carlos Augusto Monteiro.

**Data curation:** Eurídice Martínez Steele.

**Formal analysis:** Eurídice Martínez Steele.

**Writing – original draft:** Eurídice Martínez Steele, Neha Khandpur, Carlos Augusto Monteiro.

**Writing – review & editing:** Eurídice Martínez Steele, Neha Khandpur, Maria Laura da Costa Louzada, Carlos Augusto Monteiro.

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
