## [Decision Letter · Decision Letter 0]

21 May 2020

PONE-D-20-02900

Association between dietary share of ultra-processed foods and urinary concentrations of phthalates and bisphenol in a nationally representative US sample

PLOS ONE

Dear Dr. Martinez Steele,

Thank you for submitting your manuscript to PLOS ONE. After careful consideration, we feel that it has merit but does not fully meet PLOS ONE’s publication criteria as it currently stands. Therefore, we invite you to submit a revised version of the manuscript that addresses the points raised during the review process.

We look forward to receiving your revised manuscript.

Kind regards,

Leng Huat Foo, PhD

Academic Editor

PLOS ONE

Additional Editor Comments (if provided):

Please address all comments and questions raised by both the reviewers before final decision of publication can be made.

- Martínez Steele, Eurídice, and Carlos A. Monteiro. "Association between dietary share of ultra-processed foods and urinary concentrations of phytoestrogens in the US." Nutrients 9.3 (2017): 209.

 The text that needs to be addressed involves some sentences of the Abstract and of the  Discussion.

In your revision ensure you cite all your sources (including your own works), and quote or rephrase any duplicated text outside the methods section. Further consideration is dependent on these concerns being addressed.

Reviewers' comments:

Reviewer's Responses to Questions

**Comments to the Author**

1. Is the manuscript technically sound, and do the data support the conclusions?

Reviewer #1: Yes

Reviewer #2: Yes

2. Has the statistical analysis been performed appropriately and rigorously? 

Reviewer #1: Yes

Reviewer #2: Yes

3. Have the authors made all data underlying the findings in their manuscript fully available?

Reviewer #1: Yes

Reviewer #2: Yes

4. Is the manuscript presented in an intelligible fashion and written in standard English?

Reviewer #1: No

Reviewer #2: Yes

5. Review Comments to the Author

Reviewer #1: This manuscript describes the cross-sectional relationships between ultra-processed food consumption and urinary biomarkers of phthalate and bisphenol exposures in the U.S. using NHANES. The authors find that exposures to several of these chemicals are associated with greater consumption of ultra-processed foods. The methods are appropriate, the results are well-described, and the topic is of great importance given that chemicals and ultra-processed foods have both been associated with adverse health outcomes. Still, the paper mostly confirms findings from a prior study of the same hypothesis, but using a larger sample size. While some associations are now statistically significant in the larger sample, the magnitude and inference of associations is generally the same as the prior paper. Thus, the current manuscript could be strengthened by focusing more on what is new in the results rather than the confirmatory findings.

In particular, the analyses of ready-to-heat ultra-processed foods, dietary share of fat in UPF-derived TEI, processed foods, processed culinary ingredients, and trends in relationships over time were not previously assessed. Focusing the paper on these new analyses and what they add to the literature would be more useful for moving the field forward.

Other specific comments are below:

Abstract

1) Last sentence of first paragraph: Small clarification here and elsewhere, the study assesses urinary biomarkers of parent compounds, or urinary concentrations of metabolites of parent compounds.

2) The parent phthalates mentioned in the first paragraph should be linked to the metabolites in the results.

3) The findings reported here have been reported previously, consider focusing on what is new in the current paper.

Manuscript

4) In general, urinary bisphenol concentrations are parent compounds and should not be referred to as metabolites.

5) It would be useful to show the results of the analysis by cycle year in the Supplementary material given that exposure and use of different phthalates has changed over time (see Zota et al. Environ Health Perspect 2014), and because this is a major opportunity to look at temporal trends in associations that was not included in the previous NHANES paper.

6) Discussion: Findings of analyses presented in Supplementary Tables 5, 6, & 7 should be reported in the Results, not the Discussion.

Editorial: While the paper is mostly well-written, the article is in need of editing for spelling and grammar. There are many typos or misspellings, particularly in the tables/figures and supplementary material. In particular, “phthalate” is misspelled in many places (e.g., Table 2, Supplementary Table 3). There are also some incomplete sentences, and several one sentence paragraphs in the methods that could be knitted together.

Reviewer #2: This manuscript investigates potential associations between ultra-processed food intake and urinary phthalate and bisphenol A, F and S in the general US population based on the data from the NHANES surveys from 2009 to 2016. The manuscript is generally well written and easy to follow. The statistical techniques used appears to be appropriate, although some additional information regarding the regression analysis is required. I have uploaded my detailed comments on the manuscript.

6. PLOS authors have the option to publish the peer review history of their article (what does this mean?). If published, this will include your full peer review and any attached files.

Reviewer #1: No

Reviewer #2: Yes: Gurusankar Saravanabhavan

---

## [Author Response · Author response to Decision Letter 0]

2 Jul 2020

Thank you, we will do so.

- Martínez Steele, Eurídice, and Carlos A. Monteiro. "Association between dietary share of ultra-processed foods and urinary concentrations of phytoestrogens in the US." Nutrients 9.3 (2017): 209.

The text that needs to be addressed involves some sentences of the Abstract and of the Discussion.

In your revision ensure you cite all your sources (including your own works), and quote or rephrase any duplicated text outside the methods section. Further consideration is dependent on these concerns being addressed.

Thank you for highlighting this. We have reviewed the Discussion to ensure all sources are cited and have rephrased duplicated text.

Since these data were not a core part of our hypotheses or study objectives, we have chosen not to present results regarding ΣLMWP in the manuscript.

We have now added captions for Supporting Information files at the end of your manuscript and updated citations accordingly.

 

Reviewers' comments:

Reviewer #1: This manuscript describes the cross-sectional relationships between ultra-processed food consumption and urinary biomarkers of phthalate and bisphenol exposures in the U.S. using NHANES. The authors find that exposures to several of these chemicals are associated with greater consumption of ultra-processed foods. The methods are appropriate, the results are well-described, and the topic is of great importance given that chemicals and ultra-processed foods have both been associated with adverse health outcomes. Still, the paper mostly confirms findings from a prior study of the same hypothesis, but using a larger sample size. While some associations are now statistically significant in the larger sample, the magnitude and inference of associations is generally the same as the prior paper. Thus, the current manuscript could be strengthened by focusing more on what is new in the results rather than the confirmatory findings.

In particular, the analyses of ready-to-heat ultra-processed foods, dietary share of fat in UPF-derived TEI, processed foods, processed culinary ingredients, and trends in relationships over time were not previously assessed. Focusing the paper on these new analyses and what they add to the literature would be more useful for moving the field forward.

Thank you very much for these valuable suggestions.

Other specific comments are below:

Abstract

1) Last sentence of first paragraph: Small clarification here and elsewhere, the study assesses urinary biomarkers of parent compounds, or urinary concentrations of metabolites of parent compounds.

Following your suggestion, we have rephrased this sentence as follows

“The aim of this study was to examine the association between dietary contribution of ultra-processed foods and urinary biomarker concentrations of parent compounds or their metabolites.”

2) The parent phthalates mentioned in the first paragraph should be linked to the metabolites in the results.

This has been revised. The metabolites mentioned in the first paragraph are now the same as those mentioned in the results:

The aim of this study was to examine the association between dietary contribution of ultra-processed foods and urinary biomarker concentrations of parent compounds or their metabolites including Di(2-ethylhexyl) phthalate (ΣDEHP), Di-isononyl phthalate (ΣDiNP), Monocarboxynonyl phthalate (mCNP), Mono (3-carboxypropyl) phthalate (mCPP), Monobenzyl phthalate (mBzP), Bisphenol A (BPA), Bisphenol F (BPF) and Bisphenol S (BPS), in the US.

3) The findings reported here have been reported previously, consider focusing on what is new in the current paper.

As suggested, we have tried to focus more on the new findings in the manuscript.

Manuscript

4) In general, urinary bisphenol concentrations are parent compounds and should not be referred to as metabolites.

Thank you. This has been revised in the manuscript.

5) It would be useful to show the results of the analysis by cycle year in the Supplementary material given that exposure and use of different phthalates has changed over time (see Zota et al. Environ Health Perspect 2014), and because this is a major opportunity to look at temporal trends in associations that was not included in the previous NHANES paper.

As suggested, we have evaluated the associations by cycle. Exposures to phthalates and bisphenol have declined since 2009 (except BPS), probably reflecting the effect of legislative activity and the advocacy efforts of nongovernmental organizations on the use of phthalates in consumer products and consumer behavior (Zota et al. 2014). The trends of association between quintiles of UPF and phthalates and bisphenol concentrations in each cycle mirror the associations observed across all cycles.

We show these results in the Supplementary material (S6 Table) and have added in Results section:

“Though exposures to phthalates and bisphenol declined since 2009 (except for BPS) probably reflecting the effect of legislative activity and the advocacy efforts of nongovernmental organizations on the use of phthalates in consumer products and consumer behavior [43], these associations were also consistent across cycles.”

6) Discussion: Findings of analyses presented in Supplementary Tables 5, 6, & 7 should be reported in the Results, not the Discussion.

This has been revised and findings of analyses presented in Supplementary Tables 5, 6, & 7, are now reported in the Results.

Editorial: While the paper is mostly well-written, the article is in need of editing for spelling and grammar. There are many typos or misspellings, particularly in the tables/figures and supplementary material. In particular, “phthalate” is misspelled in many places (e.g., Table 2, Supplementary Table 3). There are also some incomplete sentences, and several one sentence paragraphs in the methods that could be knitted together.

The text has been reviewed for typos or misspellings and incomplete sentences. One sentence- paragraphs have been joined to the adjacent paragraphs.

 

Reviewer #2: This manuscript investigates potential associations between ultra-processed food intake and urinary phthalate and bisphenol A, F and S in the general US population based on the data from the NHANES surveys from 2009 to 2016. The manuscript is generally well written and easy to follow. The statistical techniques used appears to be appropriate, although some additional information regarding the regression analysis is required. I have uploaded my detailed comments on the manuscript.

Abstract

“The adverse nutritional composition of these products”. nutrition by definition is helpful to body. Suggest rewording

We have rephrased to “The deleterious nutrient profile of these products”. 

Introduction

“A recent inpatient ad libitum cross-over randomized controlled trial conducted by the US National Institute of Health concluded that individuals consumed 508 more kcal and gained an average of 0.8 kg of weight during the ultra-processed diet (> 80% energy from ultra-processed foods) and lost 0.9 kg during the non-ultra-processed diet”. State if this increase for per day or week?

Thank you. As suggested “508 more kcal” has been replaced by “508 more kcal/day”

“These non- chemically bond to the polymeric matrix chemicals, are known”. These chemicals are not bound to the polymer matrix chemically, and are known

Thank you. As suggested, we have replaced “These non- chemically bond to the polymeric matrix chemicals, are known” by “These chemicals are not bound to the polymer matrix chemically, and are known”

Methods

2.4 Dietary Assessment. Can you comment on why the total amount of ultra-processed food consumed (in gram or as proportion of total food) could be used to assess the association?

We used the dietary contribution of ultra-processed foods (as a percentage of total energy intake) as our main exposure variable. Ultra-processed food consumption as a proportion of total calorie intake, instead of absolute values, reduces bias introduced by non-differential calorie underreporting and adjusts for total energy intake. It is also the most frequently used indicator to assess associations across multiple health outcomes and has been used in population- based surveys and cohort studies in Latin America, Europe and Australia. 

However, to address the reviewer’s suggestion, we re-ran our analyses with ultra-processed foods expressed as a percentage of total grams of food. Results remained stable when the ultra-processed food consumption was weighted by the grams (% grams/day) instead of energy (% kcal/day). We have now added in supplementary material (S2 Table) these sensitivity analyses weighing ultra-processed food consumption by grams and also added in Results section:

“The tendencies remained stable when ultra-processed food was expressed as % of total gram intake instead of % of total energy though the association with mCNP, mCPP and BPS lost statistical significance and the association with BPA gained significance (S2 Table)”.

2.4 Dietary Assessment. I was wondering why total fat content is considered as a key determinant while recent research indicate that sugar content and to lesser extent the fat content of these ultra-processed food are risk factors for chronic illness.

In our specific analysis we are evaluating the association between ultra-processed food consumption and phthalate/bisphenol concentrations and not associations with chronic disease. As explained in the Data analysis section, previous studies have suggested that foods high in fat may be more contaminated by more lipophilic phthalates and BPA [60; 61]. In line with these hypotheses and previous literature we adjusted for total fat intake rather than added sugars.

2.5 Covariates. “BMI” Waist circumference is a better measure of overweight/obesity than BMI.

Both BMI and waist circumference are strong predictors of adiposity, incidence of chronic disease and premature mortality [Frank Hu 2008]. While waist circumference is more of an indirect measure of abdominal or central obesity, BMI is an indirect measure of overall adiposity. 

We used BMI as the surrogate measure of overall adiposity most commonly used in both children and adults.

2.6 Data Analysis. “1) crude (in ηmol/ml); 2) standardized in ηmol/g creatinine; 3) standardized in ηmol/g creatinine and adjusted for socio-demographic variables (sex, age group, race/ethnicity, ratio of family income to poverty, cycle); and 4) standardized in ηmol/g creatinine and adjusted for socio-demographic + energy intake above recommended (yes, no), BMI (categorical), physical activity (categorical) and current smoking (yes, no)”. How many covariates were included in each model and how many multiple comparisons were carried out. Can you provide the adjusted R2 for each covariate to understand the impact of these variable to the overall association?

The first two models did not adjust for any covariate, the third model adjusted for 5 covariates and the fourth model adjusted for 9 covariates.

We have revised the manuscript to ensure this is clear:

“four models were explored: 1) crude (in ηmol/ml); 2) standardized (in ηmol/g creatinine); 3) standardized (in ηmol/g creatinine) and adjusted for socio-demographic variables (sex, age group, race/ethnicity, ratio of family income to poverty, cycle); and 4) standardized (in ηmol/g creatinine) and adjusted for socio-demographic variables (sex, age group, race/ethnicity, ratio of family income to poverty, cycle), energy intake above recommended (yes, no), BMI (categorical), physical activity (categorical) and current smoking (yes, no).”

We performed tests for linear trend to assess increases or decreases of concentration levels across quintiles of the dietary share of ultra-processed foods (Table 3). We additional calculated the percent difference in urinary chemical concentrations between first and fifth quintile of the dietary share of ultra-processed foods.

As we are not testing predictive models, we consider that including the adjusted R2 for each covariate would not add much to these analyses. Covariates are included in the models to try to control for confounding rather than to increase the explicability of the model. 

2.6 Data Analysis. “On the basis of the multivariable regression models, we calculated and plotted the estimated margins for each metabolite according to quintile of relative ultra-processed food consumption”. My understanding is that this is not directly the food consumption but total energy derived from ultra-processed food consumption, is that right?

Yes, exactly. We have rephrased this sentence to ensure it is clear:

“From these regression models, we estimated: a) geometric means of urinary chemical concentrations across quintiles of the energy contribution of ultra-processed foods as e(constant + β), where β are each of the estimated regression coefficients. On the basis of the multivariable regression models, margins were estimated at the means of all covariates; b) percent difference in urinary chemical concentrations comparing first and fifth quintile of the energy contribution of ultra-processed foods as (e(β4) – 1) × 100% with 95% CIs estimated as (e(β4 ± critical value × SE) – 1) × 100%, where β4 and SE are the estimated regression coefficient and standard error for the fifth quintile, respectively”.

2.6 Data Analysis. “Previous studies have suggested that foods high in fat may be more contaminated by phthalates and BPA that are more lipophilic [60; 61]. For this reason and to test the robustness of the associations, we conducted sensitivity analyses (using the multivariable socio-demographic and life-style adjusted model) also adjusting for total fat intake (% of total energy intake) (continuous)”. Have you tested association between the % of ultra-processed food and metabolite levels? Just wondering how the strength of association would vary?

Below, we have included a table with results of the association between the dietary share of ultra-processed foods with ΣDEHP metabolites (mEHP, mEHHP, mEOHP and mECPP) and ΣDiNP metabolites (mNP/miNP, mCOP and mONP).

Table X. Adjustedc phthalate metabolite levels (standardized in ηmol/g creatinine) according to the quintiles of the dietary contribution of ultra-processed foods. Subsample of US population aged 6 + years (NHANES 2009-2016) (n=9,416)

 Quintile of dietary contribution of ultra-processed foods (% of total energy intake)a 

Parent compound Metabolite Q1 Q2 Q3 Q4 Q5 p for trend

ΣDEHP (GMb) Mono(2-ethylhexyl) phthalate; mEHP 5.6 5.8 5.2 5.3 5.2 0.005

 Mono(2-ethyl-5-hydroxyhexyl) phthalate; mEHHP 28.4 27.6 26.3 27.2 27.2 0.2

 Mono(2-ethyl-5-oxohexyl) phthalate; mEOHP 17.8 17.7 16.9 17.7 17.6 0.7

 Mono(2-ethyl-5-carboxypentyl) phthalate; mECPP 42.7 41.7 40.0 42.0 42.7 0.9

ΣDiNP (GM) Mono-isononyl phthalate; mNP/miNP 4.1 4.4 3.9 4.4 4.2 0.8

 Monocarboxyoctyl phthalate; mCOP 40.4 45.5 44.7 51.6 51.4 <0.001

aMean (range) dietary share of ultra-processed foods per quintile: 1st=27.1 (0 to 39.5); 2nd= 46.8 (39.5 to 53.3); 3rd= 59.3 (53.3 to 65.2); 4th=71.1 (65.2 to 77.7); 5th= 87.3 (77.7 to 100)

bGM= geometric mean

cAdjusted for cycle, sex, age group, race/ethnicity, and income, in addition to energy intake above recommended levels (y/n), BMI (underweight, normoweight, overweight, obesity), physical activity (low, medium, high) and current smoking (y/n).

Results

For some metabolites, concentration levels varied according to total energy intake above recommended levels (positive association for ΣDEHP, mCNP and mCPP) and total fat above recommended levels (positive association for ΣDiNP and mCNP, and inverse for BPS), and fat in ultra-processed foods derived total energy intake (positive association for all metabolites, except for BPS which was inverse and ΣDEHP with no association). Can a similar analysis be carried out for total sugar intake via ultra-processed food?

Our hypotheses to test dietary share of total fat in UPF derived total energy intake as exposure variable was informed by published literature. In this study we did not carry out analysis using quintile of dietary share of added sugars in UPF derived total energy intake as exposure variable (as we did with total fat in S3 Table) because there is no published evidence that we are aware of, suggesting foods high in added sugars may be more contaminated by phthalates and BP.

Discussion

The sentence appears to assume that the association between MBzP, BPF and BPS are casual. Suggest replacing "failed to observe" with "did not observe".

Following your suggestion, we have replaced "failed to observe" with "did not observe".

What is PC?

We have substituted “PC” by “polycarbonate” in the manuscript.

additional pathway? In medical literature, it is well-known that increased calorie consumption per day is a risk factor for various health outcomes including obesity.

Following your suggestion, we have replaced "alternative pathway" with "additional pathway".

Table 3

What is the coefficient of determination for this and other models listed in this table?

As we are not testing predictive models, we consider that including the adjusted R2 for each covariate would not add much to these analyses. Covariates are included in the models to try to control for confounding rather than to increase the explicability of the model.

---

## [Decision Letter · Decision Letter 1]

9 Jul 2020

PONE-D-20-02900R1

Association between dietary contribution of ultra-processed foods and urinary concentrations of phthalates and bisphenol in a nationally representative US sample

PLOS ONE

Dear Dr. Martinez Steele,

Thank you for submitting your manuscript to PLOS ONE. After careful consideration, we feel that it has merit but does not fully meet PLOS ONE’s publication criteria as it currently stands. Therefore, we invite you to submit a revised version of the manuscript that addresses the points raised during the review process.

We look forward to receiving your revised manuscript.

Kind regards,

Leng Huat Foo, PhD

Academic Editor

PLOS ONE

Additional Editor Comments (if provided):

Many thanks for your kind consideration to address all comments and questions raised. However, the title of the manuscript submitted requires some minor changes. It is too general to state "US sample", instead of the "US population aged 6 years and older" in the title of the manuscript.

Reviewers' comments:

Reviewer's Responses to Questions

**Comments to the Author**

1. If the authors have adequately addressed your comments raised in a previous round of review and you feel that this manuscript is now acceptable for publication, you may indicate that here to bypass the “Comments to the Author” section, enter your conflict of interest statement in the “Confidential to Editor” section, and submit your "Accept" recommendation.

Reviewer #1: All comments have been addressed

2. Is the manuscript technically sound, and do the data support the conclusions?

Reviewer #1: Yes

3. Has the statistical analysis been performed appropriately and rigorously? 

Reviewer #1: Yes

4. Have the authors made all data underlying the findings in their manuscript fully available?

Reviewer #1: Yes

5. Is the manuscript presented in an intelligible fashion and written in standard English?

Reviewer #1: Yes

6. Review Comments to the Author

Reviewer #1: (No Response)

7. PLOS authors have the option to publish the peer review history of their article (what does this mean?). If published, this will include your full peer review and any attached files.

Reviewer #1: No

---

## [Author Response · Author response to Decision Letter 1]

10 Jul 2020

We thank you for the suggestion. We have changed the title as suggested and now reads:

“Association between dietary contribution of ultra-processed foods and urinary concentrations of phthalates and bisphenol in a nationally representative sample of the US population aged 6 years and older”.

---

## [Editor Report · Decision Letter 2]

14 Jul 2020

Association between dietary contribution of ultra-processed foods and urinary concentrations of phthalates and bisphenol in a nationally representative sample of the US population aged 6 years and older

PONE-D-20-02900R2

Dear Dr. Martinez Steele,

We’re pleased to inform you that your manuscript has been judged scientifically suitable for publication and will be formally accepted for publication once it meets all outstanding technical requirements.

Kind regards,

Leng Huat Foo, PhD

Academic Editor

PLOS ONE
---

## [Editor Report · Acceptance letter]

15 Jul 2020

PONE-D-20-02900R2 

Association between dietary contribution of ultra-processed foods and urinary concentrations of phthalates and bisphenol in a nationally representative sample of the US population aged 6 years and older 

Dear Dr. Martínez Steele:

I'm pleased to inform you that your manuscript has been deemed suitable for publication in PLOS ONE. Congratulations! Your manuscript is now with our production department. 

Kind regards, 

on behalf of

Dr. Leng Huat Foo 

Academic Editor

PLOS ONE